# The Synergistic Effect of Compound Sugar with Different Glycemic Indices Combined with Creatine on Exercise-Related Fatigue in Mice

**DOI:** 10.3390/foods13030489

**Published:** 2024-02-03

**Authors:** Hui Liao, Song Zhu, Yue Li, Dejian Huang

**Affiliations:** 1State Key Laboratory of Food Science and Resources, Jiangnan University, Wuxi 214122, China; 2School of Food Science and Technology, Jiangnan University, Wuxi 214122, China; 3Department of Food Science and Technology, National University of Singapore, Singapore 117542, Singapore

**Keywords:** compound sugar, creatine, glycemic response, emulsion, exercise-related fatigue

## Abstract

In this study, a compound sugar (CS) with different glycemic index sugars was formulated via hydrolysis characteristics and postprandial glycemic response, and the impact of CS and creatine emulsion on exercise-related fatigue in mice was investigated. Thirty-five C57BL/6 mice were randomly divided into five groups to supply different emulsions for 4 weeks: initial emulsion (Con), glucose emulsion (62 mg/10 g MW glucose; Glu), CS emulsion (62 mg/10 g MW compound sugar; CS), creatine emulsion (6 mg/10 g MW creatine; Cr), and CS and creatine emulsion (62 mg/10 g MW compound sugar, 6 mg/10 g MW creatine, CS-Cr). Then, the exhaustion time of weight-bearing swimming and forelimb grip strength were measured to evaluate the exercise capacity of mice, and some fatigue-related biochemical indexes of blood were determined. The results demonstrated that the ingestion of CS significantly reduced the peak of postprandial blood glucose levels and prolonged the energy supply of mice compared to ingesting an equal amount of glucose. Mouse exhaustion time was 1.22-fold longer in the CS group than in the glucose group. Additionally, the supplementation of CS increased the liver glycogen content and total antioxidant capacity of mice. Moreover, the combined supplementation of CS and creatine increased relative forelimb grip strength and decreased blood creatine kinase activity. The findings suggested that the intake of CS could enhance exercise capacity, and the combined supplementation of CS and creatine has a synergistic effect in improving performance.

## 1. Introduction

In the process of fitness or exercise, a lack of energy substances can result in a decrease in the performance of muscles [1]. Additionally, exercise-related fatigue can be caused by the accumulation of some particular metabolites, such as lactate, which also can impair the function of muscles [2]. During training and contests, numerous athletes and amateurs often opt for sports nutritional supplements to enhance their energy levels, prevent the buildup of adverse metabolites, and enhance their athletic capabilities.

Modern sports nutrition strategies are based on the principle that adequate carbohydrates should be provided to support the physical activity of the body [3]. Carbohydrate supplementation can increase pre-exercise muscle glycogen storage, as well as enhance endurance and team exercise performance [4]. Furthermore, the glycemic index (GI) of carbohydrates may influence endurance during exercise. It was reported that substituting sucrose with isomaltose (a low GI disaccharide) significantly increases the exercise endurance of mice for four weeks [5]. However, a few researchers have argued that low GI carbohydrates do not have a noticeable impact on exercise capacity [6,7]. This may be due to the slow release of energy from low GI carbohydrates, which may not adequately fuel the body during exercise. In light of this, we propose that using a combination of carbohydrates with different GI values could be a more effective strategy to relieve exercise-related fatigue and enhance exercise endurance. This strategy could not only reduce postprandial hyperglycemia caused by high GI sugars but also properly maintain blood sugar levels and carbohydrate availability during exercise. Currently, research on relieving exercise-related fatigue primarily focuses on low GI carbohydrates. However, with limited reports, supplements with different GI sugars are rarely mentioned. Therefore, it is worthwhile to explore the effects of compound sugars composed of various GI sugars on exercise fatigue.

Creatine is capable of synthesizing phosphocreatine, which helps maintain ATP levels. Prior research has reported that creatine supplements can enhance muscle strength and explosive power during exercise [8]. Furthermore, several studies have found that supplementing with creatine can boost glucose absorption and increase the glycogen content in both human and animal skeletal muscle [9]. According to Ceddia et al., supplementing with creatine might increase glucose oxidation in L6 rat skeletal muscle cells while reducing lactate production [10]. Many individuals frequently combine multiple sports nutrients to maximize the benefits and improve their exercise performance. It was found that combining carbohydrate and creatine ingestion could improve creatine retention in the body compared to supplying creatine alone, leading to increased power output during the later stages of a bicycle time trial sprint [11,12]. Nevertheless, the effects of combining carbohydrate and creatine ingestion on exercise performance remain inconsistent. Several studies indicate that it does not significantly improve athletes’ anaerobic exercise capacity compared to supplying creatine alone [13]. It is worth noting that carbohydrates used in these studies are single sugars because they can be quickly absorbed by the body. The effects of compound sugar with different GI combined with creatine supplementation on exercise performance need to be further investigated.

In this study, the hydrolysis characteristics and postprandial glycemic responses of several different GI sugars were determined, and a compound sugar (CS) suitable for exercise ingestion was formulated. Furthermore, we prepared an emulsion by homogenizing the mixture of whey protein isolate (WPI) solution and linolenic acid using the high-pressure homogenizer. CS and creatine were added to the WPI solution. The effects of CS combined with creatine emulsion supplementation on the exercise capacity of mice trained for 4 weeks were studied by measuring the exhaustion time of weight-bearing swimming and forelimb grip strength. Via the determination of glycogen contents and biochemical indexes in the blood, such as lactate, blood urea nitrogen (BUN), creatinine, creatine kinase (CK), and total antioxidant capacity (T-AOC), the anti-fatigue effects of CS and creatine were compared. The obtained results will provide a scientific basis for the compounding supplement of sugars with different GI and creatine to improve exercise performance and relieve exercise-related fatigue.

## 2. Materials and Methods

### 2.1. Reagents and Animals

Glucose, sucrose, and maltodextrin were purchased from Xiwang Foodstuffs Co., Ltd. (Jinan, China). Isomaltulose was bought from Harnikon Biotechnology Co., Ltd. (Binzhou, China). All other reagents were provided by Abbott Trading Co., Ltd. (Shanghai, China). Kits for D-glucose, glycogen content, lactate, BUN, creatinine, CK, and T-AOC were provided by Nanjing Jiancheng Bioengineering Institute (Nanjing, China). All other chemicals used were of analytical reagent grade.

Male C57BL/6 mice (aged, 6 weeks; body weight, 20 ± 2 g) were purchased from Sipeifu Biotechnology Company (Beijing, China). The mice were housed in separate cages in a specific pathogen-free environment for adaptive rearing for the first week where they were provided with clean water and basic feed. Each experiment was approved by the Ethics Committee of the Laboratory Animal Center of Jiangnan University (Permission number: JN.No20220415c0500620[128]), which was conducted by the Guide for the Care and Use of Laboratory Animals (8th edition, National Academy of Sciences Press).

### 2.2. Emulsion Preparation

The preparation method of the emulsion was referred to in a previous study with some modifications [14]. The WPI solution with a concentration of 10 mg/mL was prepared and mixed with various compounds (i.e., 250 mg/mL glucose, 250 mg/mL CS, and 25 mg/mL creatine) [7,15]. CS was composed of glucose, maltodextrin, sucrose, and isomaltose with a mass ratio of 4:5:6:15. Then, the solution was mixed with linolenic acid at a ratio of 9:1 (*v*/*v*). The mixture was treated using a high-speed dispersing and emulsifying unit (IKA-ULTRA-TURRAX T25 basic, IKA Works, Inc., Wilmington, USA) at a high speed of 20,000 r/min for 2 min to obtain a coarse emulsion. The coarse emulsion was further homogenized using a homogenizer (Panda PLUS 2000; GEA NiroSoavi, Inc., Parma, Italy) at 40 MPa 2 times at 25 °C. The obtained emulsion was used for mice experiments.

### 2.3. Hydrolytic Properties of Mice Small Intestinal Extract

The mice’s small intestinal extract (MSIE) was obtained by referring to the method previously described [16]. Firstly, the healthy mice that did not receive any treatment were killed via neck dissection after 16 h of fasting without water, and the small intestinal parts were dissected immediately. After gently squeezing to exclude the contents, it was rinsed several times with cold saline, blotted dry with filter paper, weighed, and clipped. Then, it was homogenized with phosphate buffer (pH 6.8, 0.1 M, 1:10 dilution; *w*/*v*), centrifuged (4 °C, 4000 r/min, 10 min), and the supernatant was aspirated to obtain the MSIE. To determine the α-glucosidase activity of the MSIE, different disaccharides (maltose (α-1,4), sucrose (α-1,2), and isomaltulose (α-1,6), 1%, *w*/*v*, 500 μL) were added to MSIE (500 μL) at 37 °C for 15 min. After the reaction, individual samples were boiled for 2 min. The amount of glucose released was measured using a D-glucose assay kit.

The hydrolytic properties were determined by referring to previous studies, with some modifications [17]. Generally, 500 μL of different substrate solutions (1 g/L) were prepared using phosphate buffer (pH 6.8; 0.1 M) and then combined with the MSIE (500 μL). In addition, to inhibit microorganisms from utilizing the glucose released during incubation, 1 μL of ampicillin (5 g/L) was added. The reaction was conducted at 37 °C, and hydrolysates were collected at 0, 0.5, 1, 2, 4, 8, 12, and 24 h. Using a D-glucose assay kit, the amount of released glucose was measured, and the hydrolysis ratio of each sugar was calculated.

### 2.4. Glycemic Responses after the Ingestion of Different Sugars in a Mouse Model

The postprandial glycemic responses were carried out by referring to a previous study [18]. Following a period of fasting without water for 16 h, mice were administered a 250 μL dose of various sugar solutions (15%, *n* = 7) using a gavage needle. Blood was collected at 0, 15, 30, 60, 90, and 120 min from the mice’s tail veins. The glucose level was measured using a blood glucose meter (Yuwell 580).

### 2.5. Mice Grouping and Training Program

The mice were randomly divided into five groups (*n* = 7) to supply different emulsions: (1) the blank control group was treated with initial emulsion (Con); (2) the glucose control group was treated with glucose emulsion (62 mg/10 g MW glucose; Glu); (3) the CS group was treated with CS emulsion (62 mg/10 g MW CS); (4) the creatine group was treated with creatine emulsion (6 mg/10 g MW creatine; Cr); (5) and the CS and creatine combination group was treated with CS and creatine emulsion (62 mg/10 g MW CS, 6 mg/10 g MW creatine, CS-Cr). After each treatment, all mice were given a rest period of 5 min before they were subjected to continuous swimming for 20 min. The depth of the water used for swimming was approximately 20 cm and maintained within a temperature range of 28 ± 1 °C. This routine was maintained consistently for each group, six times per week, for four weeks [19].

### 2.6. Weight-Bearing Swimming Test

After 4 weeks of nutritional supplementation, the exhaustion time of weight-bearing swimming in mice was determined by referring to previous reports, with some modifications [19]. Each mouse’s tail was tied to a 5% body weight lead block, and then it was assessed in a swimming tank. Then, the time from entering the water till exhaustion was recorded. Mice were deemed to have reached exhaustion if they failed to surface within 5 s of sinking underwater.

### 2.7. Forelimb Grip Strength Test

Following the four-week swimming training, the grip strength of the mice’s forelimbs was measured using a grip strength detector (BIO-GS3, Bioseb, Inc., Paris, France). The data were recorded by taking the maximum value three times in a row. The following formula was used to determine the relative grip strength:(1)$$\mathrm{Relative}\;\mathrm{Grip}\;\mathrm{strength}\;=\frac{\mathrm{Grip}\;\mathrm{strength}}{\mathrm{Body}\;\mathrm{weight}}$$

### 2.8. Determination of Biochemical Parameters

At the end of the experiment, the mice were sacrificed for the collection of blood samples from the orbital plexus under ether anesthetization, and the blood samples were centrifuged at 1500× *g* for 15 min to take the serum. The liver and muscle samples were rinsed in saline and placed in liquid nitrogen for rapid freezing. Then, the serum and tissue samples were placed in a −80 °C refrigerator until tested. The glycogen content, lactate, BUN, creatinine, CK level, and T-AOC were all determined using corresponding kits.

### 2.9. Statistical Analysis

Data were expressed as mean ± standard deviation (SD). The results of hydrolytic properties and glycemic responses were analyzed via repeated measures analysis of variance (ANOVA) with SPSS (version 20.0; SPSS, Chicago, IL, USA), and the other data were analyzed via one-way ANOVA. Duncan’s test was used to measure the difference between different treatments. *p* < 0.05 was taken as the standard of statistical significance.

## 3. Results and Discussion

### 3.1. Hydrolytic Properties of Mammalian Glucosidase on Different Glycosidic-Linked Sugars

MSIE was obtained from the small intestines of mice, which have been proven to possess α-1,4, α-1,2, and α-1,6 hydrolytic activities [20]. Therefore, the use of MSIE provided a more precise reflection of the hydrolysis of different glycosidic linkage sugars in humans than the use of α-glucosidase derived from fungi. The enzyme activities of the MSIE were evaluated using various types of substrates comparable (Table 1). In mice, maltase exhibited the greatest activity among the three types of disaccharidases, followed by sucrase and palatinase. The activity of maltase was 6.47 and 18.33 times higher than that of sucrase and palatinase, respectively. Maltase activity was 6.47 and 18.33 times greater than the activity of sucrase and palatinase, respectively. In humans, the relative activity of sucrose and palatinase compared to maltase was generally comparable [16].

The main digestion and absorption sites of disaccharides and oligosaccharides are in the small intestine [21]. To determine the hydrolytic properties, MISE was used to analyze the amounts of glucose released from various sugars (Figure 1). The results showed that maltose, maltodextrin, and sucrose were almost entirely hydrolyzed over 24 h. A total of 80% of isomaltulose was hydrolyzed into glucose and fructose. Maltose and maltodextrin were quickly converted to glucose via hydrolysis, while sucrose was hydrolyzed at a slower pace. The hydrolysis of isomaltulose occurred at the slowest rate. Previous studies on the in vitro digestion of isomaltulose via mammalian intestinal α-glucosidase (including humans) have similar results [22]. The primary reason for this finding is that the enzymes that function in hydrolyzing the α-1,4 glycosidic bond are relatively abundant in MSIE compared to the others.

### 3.2. Glycemic Responses after the Ingestion of Individual Sugars in a Mouse Model

The glycemic responses of mice after the ingestion of glucose, sucrose, isomaltose, and maltodextrin (Figure 2) were similar to the hydrolytic rate in vitro. The change in glycemic level after ingesting a certain amount of glucose reached the maximum level 15 min after ingestion, and then the blood glucose concentration decreased sharply with the prolongation of post-ingestion time. Maltodextrin significantly reduced the glycemic responses compared to glucose, while the change also reached its highest level 15 min after ingestion and then decreased rapidly. The highest blood glucose was observed 30 min after sucrose ingestion, and there was no significant difference in the concentration from 15 to 30 min. Isomaltulose could be entirely digested and absorbed in the small intestine [23]. In this study, the postprandial glycemic response of isomaltulose intake was significantly lower than that of sucrose. The blood glucose concentration reached the peak value of 60 min after ingestion, and there was no significant difference between 15 and 60 min. These results were comparable to those of a previous human experiment that isomaltulose reduced the increase in postprandial blood glucose and insulin concentration compared with sucrose [23]. The slow hydrolysis and the stable postprandial glycemic responses of isomaltulose have prompted research into its benefits for exercise performance [24].

### 3.3. Glycemic Responses after the Ingestion of CS in a Mouse Model

Studies have demonstrated that the ratio of fructose to glucose of (0.5~1): 1 could promote exogenous carbohydrate oxidation and gastrointestinal comfort [25]. In our study, the sources of fructose were sucrose and isomaltulose; all four sugars provided sources of glucose. The formulation of CS was established by considering the hydrolysis properties in vitro and postprandial glycemic responses of each sugar. The formulation was found to have a refreshing taste suitable for athletic supplementation via pre-experiment sensory evaluation. Among them, sucrose was included within the CS formulation to maintain sweetness, with isomaltulose added to prolong the energy supply, and glucose and maltodextrin were included for the energy supply rapidly.

Figure 3 depicts the postprandial glycemic glucose response following the ingestion of CS. Within 15 min of CS ingestion, blood glucose concentration reached its peak, which remained relatively stable from 15 to 30 min. It indicated that CS may ensure a timely and reliable supply of energy prompt and consistent energy supply. In comparison to ingesting an equal amount of glucose, the consumption of CS resulted in significantly reduced blood glucose peaks and more gradual changes in blood glucose levels. Beyond the 30 min mark, the average blood glucose level of mice who consumed CS remained consistently higher than those who consumed glucose throughout the experiment, with the gap widening over time. The results showed that CS could achieve better maintenance of glucose concentrations and provide more stable carbohydrate availability during exercise compared to glucose [7].

### 3.4. Body Weight and Exercise Performance of Mice

The body weight of the mice during exercise training is presented in Table 2. With time, the body weight in each group showed an increase, but there were no significant differences between the five groups.

We evaluated the exercise behavior of mice after a 4-week treatment period to explore the effects of CS and creatine on their exercise capacity, using the weight-bearing swimming and forelimb grip strength tests. The exhaustive swimming time was measured to determine their endurance, and the results are presented in Figure 4. There were no significant differences between the Glu group and the Con group. However, compared to those in the glucose group, mice supplemented with CS demonstrated a significant increase in their average exhaustion time compared to glucose (The exhaustion time of CS and CS-Cr was 1.22 and 1.18 times that of Glu, respectively). Moreover, there was no significant difference between CS and CS-Cr. These results demonstrated a significant improvement in mice’s exercise endurance when supplied with CS.

The forelimb grip of mice is shown in Figure 5A, and there was no significant difference among groups. Relative grip strength, the ratio of mouse grip strength to body weight, holds particular relevance in certain sports like weightlifting, where athletes’ weight must be considered [26]. The relative grip strength of the mice supplemented with creatine or CS was significantly higher compared to the glucose group (Figure 5B). While the separate supplementation of creatine and CS had no significant impact on the relative grip strength of mice, the combination of both significantly enhanced their relative grip strength. This outcome may be attributed to the potential of carbohydrates and creatine co-ingestion to facilitate the uptake and retention of creatine by muscle cells, thereby enhancing muscle strength [11].

### 3.5. Glycogen Contents of Mice

Energy consumption during exercise could result in muscle fatigue. During exercise, skeletal muscle utilizes carbohydrates or fatty acids as sources of fuel. Specifically, in the case of prolonged exercise, the availability of glycogen content is considered to be a critical limiting factor for energy utilization [27]. Consequently, the abundant storage of endogenous glycogen plays a crucial role in facilitating exercise capacity.

The glycogen contents of mice in each group are presented in Figure 6. The muscle glycogen content of the mice supplemented with glucose, creatine, and CS was significantly higher compared to the blank control group, with no significant difference among the four groups (Figure 6A). The liver glycogen content in the CS group was significantly higher than that observed in the glucose control group (Figure 6B). Compared with the blank control group, supplementation with CS and creatine alone led to increased liver glycogen content in mice, although the difference was not statistically significant. However, the liver glycogen content in the CS-Cr group was significantly higher compared to the blank control group, with a 1.31-fold increase observed. These results suggested that CS supplementation could raise the content of glycogen in mice, and combined supplementation with CS and creatine could additionally enhance liver glycogen content.

### 3.6. Serum Fatigue Indexes in Mice

It is widely acknowledged that muscle fatigue plays a crucial role in the fluctuations of sports performance [28]. Several biochemical indicators, such as lactate, BUN, creatinine level, and CK activity, were assessed to investigate the metabolic alterations related to the anti-fatigue effect of CS and creatine.

Anaerobic glycolysis during exercise leads to lactate production, and the excessive buildup of lactate in the body contributes to localized muscle soreness and detrimentally impacts sports performance [29]. Figure 7A illustrates that no significant variation in lactate content was observed among the different groups. This suggested that neither CS nor creatine could prevent muscle injury by reducing the production of lactate during anaerobic exercise [30]. BUN level serves as an additional fatigue index produced from protein and amino acid catabolism within the body. During high-intensity exercise, the body metabolizes protein as a result of insufficient energy from fats and carbohydrates, consequently elevating BUN levels [31]. Figure 7B demonstrates that no significant difference in BUN content existed between each experimental group and the blank control group.

The energy is necessary for metabolic activities and is derived from ATP, which is subsequently converted into ADP. To ensure a continuous supply of ATP during exercise, the energy released by creatine phosphate can be utilized for ATP synthesis. Creatinine is the byproduct of creatine phosphate metabolism [32]. The creatine phosphate content in mice could be indirectly assessed by measuring the creatinine concentration in their blood. In this experiment, mice supplemented with CS and creatine displayed a significant increase in serum creatinine content (Figure 7C). This can likely be attributed to the combined intake of CS and creatine, which prompts the body to retain higher levels of creatine to elevate creatinine production.

CK is abundantly present in muscle cells, especially in skeletal muscle [33]. An elevation in blood CK levels typically indicates muscle cell damage and leakage of CK into the bloodstream [34]. Strenuous exercise is commonly associated with muscle damage and an increase in blood CK activity [35]. Notably, the CK activity observed in the CS group was significantly lower than that in the glucose control group (Figure 7D). While the creatine group exhibited a slight decrease in CK activity compared to the blank control group, no significant difference was observed. However, the combined supplementation of creatine and CS resulted in a further reduction in CK activity, which was significantly different from that of the control group. These findings suggested that CS intake alone or in combination with creatine can serve as preventive measures against muscle injury induced by excessive exercise, thereby enhancing exercise performance.

### 3.7. Total Antioxidant Capacity of Mice

Oxidative stress caused by strenuous exercise will also lead to body fatigue and therefore affect sports performance. Prior studies have illustrated that consistent physical exercise can improve T-AOC of the body and reduce lipid peroxidation [36]. In this study, the T-AOC of mice in the Cr group was significantly higher than that of mice in other groups (Figure 8). Additionally, Sestili’s research suggested that creatine can provide cytoprotection in mammalian cells when subject to oxidative injury via direct antioxidant activity [37]. The T-AOC of mice in the CS group and the CS-Cr group were significantly higher in comparison to the control group, and the T-AOC of the CS-Cr group was even more than that of the CS group. Based on these findings, the supplementation of creatine and CS could effectively augment the antioxidant capacity of mice, thereby reducing exercise-related fatigue and boosting exercise performance.

## 4. Conclusions

In this study, we formulated a compound sugar and prepared an emulsion by homogenizing the mixture of WPI solution and linolenic acid. The combined effects of CS with creatine emulsion supplementation on exercise-related fatigue in mice were also evaluated. The CS was formulated as the weight ratio of glucose/maltodextrin/sucrose/isomaltose at 4:5:6:15. The ingestion of CS reduced the peak of postprandial blood glucose levels and prolonged the energy supply. The combination of CS and creatine demonstrated superior benefits in terms of exhaustion time, forelimb grip strength, glycogen content, and CK activity of mice, compared to either CS or creatine alone. Additionally, both creatine and CS increased the T-AOC. In summary, the combined supplementation of CS and creatine exhibited a synergistic effect to enhance exercise capacity and relieve exercise-related fatigue in mice. Further studies are required to investigate the effects of CS-Cr compared to glucose or low GI sugars (i.e., isomaltulose) with creatine. Moreover, the animal experimental results need to be further verified by randomized controlled trials.

## Figures and Tables

**Figure 1 foods-13-00489-f001:**
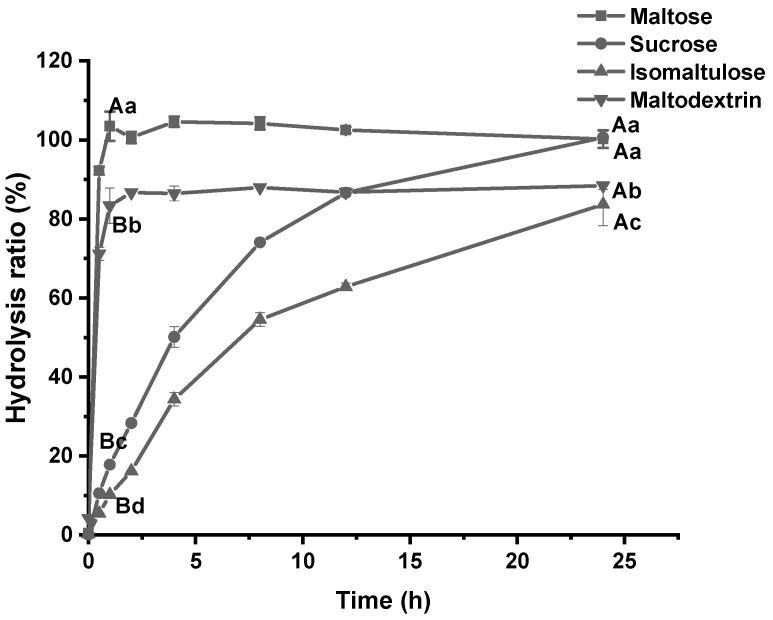
Hydrolytic properties of different types of sugar (1 g/L) for 24 h by mice small intestine extract as a source of mammalian α-glucosidases. Data are expressed as means ± SD (*n* = 3). Different capital letters indicate significant differences between time points in the same group, and different lowercase letters indicate significant differences between groups at the same time point (*p* < 0.05).

**Figure 2 foods-13-00489-f002:**
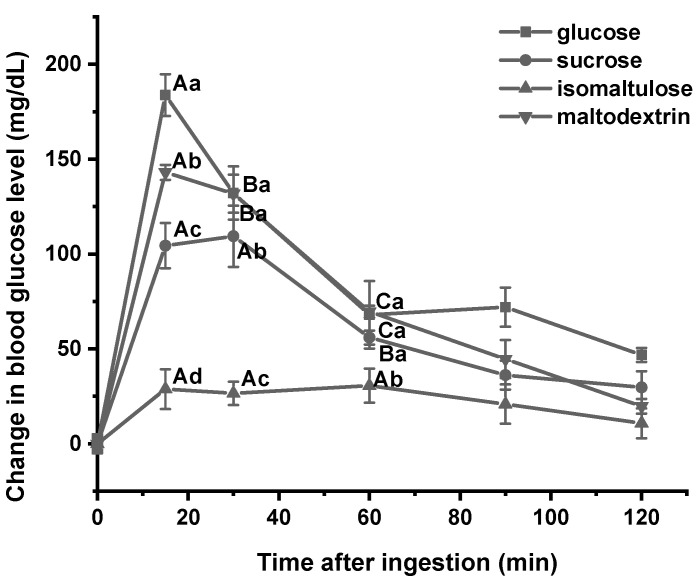
Glycemic responses after the ingestion of individual sugars. Data are expressed as means ± SD (*n* = 7). Different capital letters indicate significant differences between time points in the same group, and different lowercase letters indicate significant differences between groups at the same time point (*p* < 0.05).

**Figure 3 foods-13-00489-f003:**
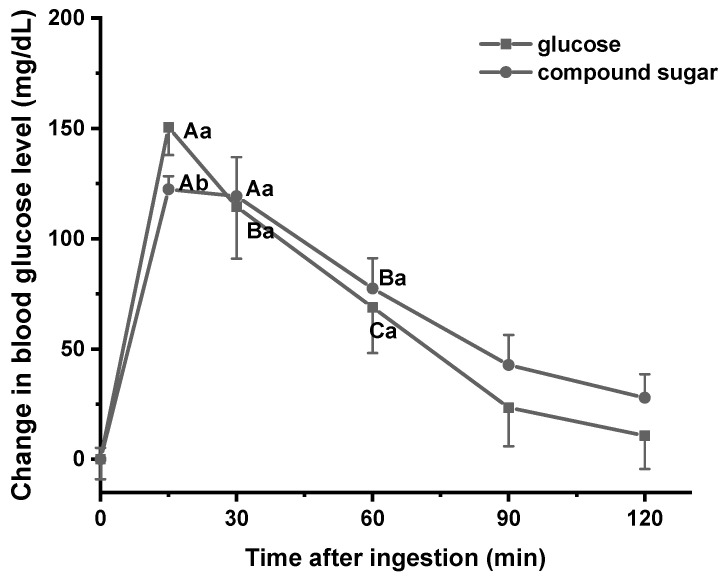
Glycemic responses after the ingestion of compound sugar. Data are expressed as means ± SD (*n* = 7). Different capital letters indicate significant differences between time points in the same group, and different lowercase letters indicate significant differences between groups at the same time point (*p* < 0.05).

**Figure 4 foods-13-00489-f004:**
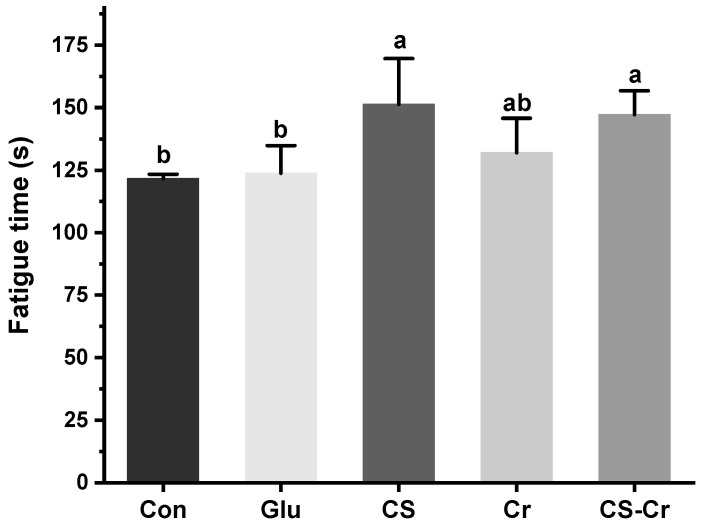
Exhaustion time of weight-bearing swimming in mice. Data are expressed as means ± SD (*n* = 7). Values with different superscripts indicate significant differences (*p* < 0.05). Con, blank control; Glu, glucose control; CS, compound sugar; Cr, creatine; CS-Cr, compound sugar and creatine.

**Figure 5 foods-13-00489-f005:**
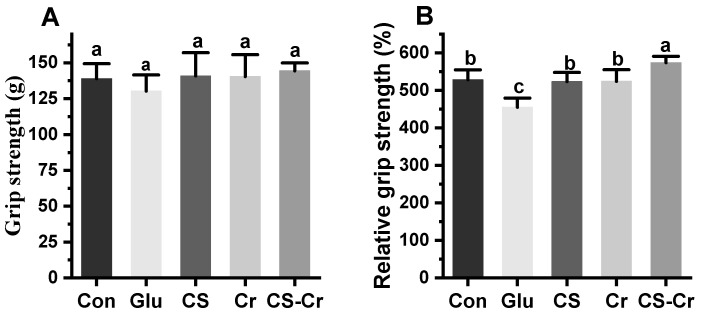
The grip strength (**A**) and relative grip strength (**B**) of mice forelimbs. Data are expressed as means ± SD (*n* = 7). Values with different superscripts indicate significant differences (*p* < 0.05). Con, blank control; Glu, glucose control; CS, compound sugar; Cr, creatine; CS-Cr, compound sugar and creatine.

**Figure 6 foods-13-00489-f006:**
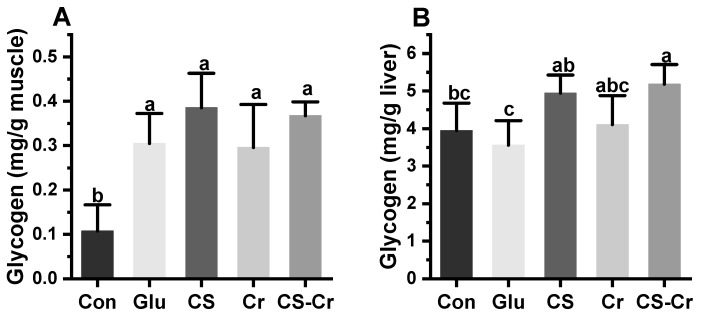
Glycogen contents after weight-bearing swimming in mice. (**A**) muscle glycogen and (**B**) liver glycogen. Data are expressed as means ± SD (*n* = 7). Values with different superscripts indicate significant differences (*p* < 0.05). Con, blank control; Glu, glucose control; CS, compound sugar; Cr, creatine; CS-Cr, compound sugar and creatine.

**Figure 7 foods-13-00489-f007:**
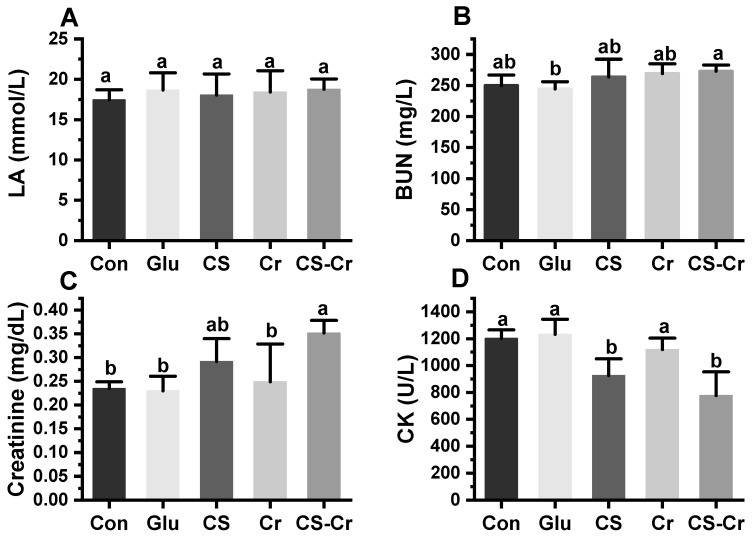
Serum fatigue indexes after weight-bearing swimming in mice. (**A**) Lactate (LA). (**B**) Blood urea nitrogen (BUN). (**C**) Creatinine. (**D**) Creatine kinase (CK). Data are expressed as means ± SD (*n* = 7). Values with different superscripts indicate significant differences (*p* < 0.05). Con, blank control; Glu, glucose control; CS, compound sugar; Cr, creatine; CS-Cr, compound sugar and creatine.

**Figure 8 foods-13-00489-f008:**
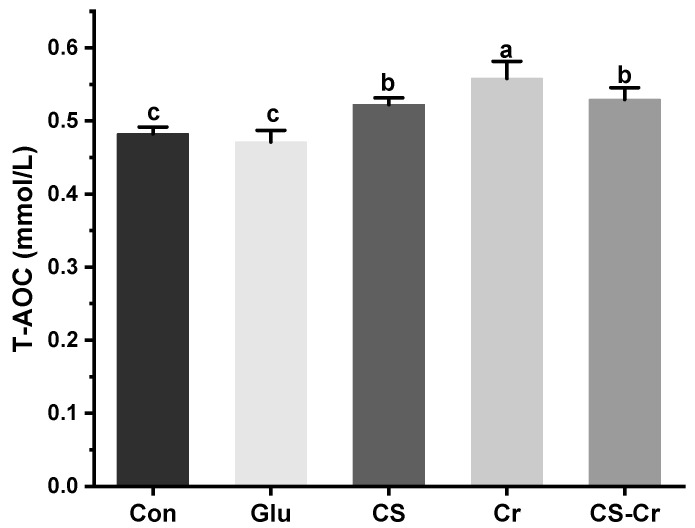
Total antioxidant capacity (T-AOC) of mice. Data are expressed as means ± SD (*n* = 7). Values with different superscripts indicate significant differences (*p* < 0.05). Con, blank control; Glu, glucose control; CS, compound sugar; Cr, creatine; CS-Cr, compound sugar and creatine.

**Table 1 foods-13-00489-t001:** Enzyme activities (U/mL) of α-glucosidases on different types of glycemic disaccharides from mice small intestine extract (MSIE).

	Maltase (α-1,4)	Sucrase (α-1,2)	Palatinase (α-1,6)
Enzyme activities (U/mL) *	1.10 ± 0.05	0.17 ± 0.09	0.06 ± 0.01

* One unit (U) is defined as the amount of substrate hydrolyzed (μmol) per minute. Data are expressed as means ± SD (*n* = 3).

**Table 2 foods-13-00489-t002:** Effects of 4-week CS and Cr supplementation on the body weight (BW) in mice.

	0 d	7 d	14 d	21 d	28 d
Con	23.57 ± 1.46	24.61 ± 1.37	25.40 ± 1.46	25.36 ± 1.70	26.14 ± 2.32
Glu	23.65 ± 1.10	24.38 ± 1.09	25.37 ± 1.38	25.55 ± 1.53	25.92 ± 1.71
CS	23.81 ± 1.74	24.31 ± 2.16	25.18 ± 2.61	25.26 ± 2.58	26.21 ± 2.44
Cr	23.15 ± 1.37	24.36 ± 1.57	25.13 ± 1.34	25.52 ± 1.36	26.14 ± 1.25
CS-Cr	23.88 ± 0.63	24.65 ± 0.95	25.46 ± 1.56	25.02 ± 1.76	26.03 ± 1.99

Data are expressed as means ± SD (*n* = 7). Con, blank control; Glu, glucose control; CS, compound sugar; Cr, creatine; CS-Cr, compound sugar and creatine.

## Data Availability

Data are contained within the article.

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
