# Peer review of "The Synergistic Effect of Compound Sugar with Different Glycemic Indices Combined with Creatine on Exercise-Related Fatigue in Mice"

_foods, 2024, doi:10.3390/foods13030489_

Round 1

Reviewer 1 Report

Comments and Suggestions for Authors

Thank you for submitting the manuscript "The Synergistic Effect of Compound Sugar with Different Glycemic Index Combined with Creatine on Exercise-Related Fatigue in Mice" to Foods. Although the idea of the study is interesting, I was unsure of the need for the study given that creatine, for example, has already been used in clinical practice for athletes. Therefore, what is the innovation of carrying out a study like this with an animal model instead of an RCT? Another important issue is that authors must provide sufficient information for the experiment to be reproduced, however, there are several gaps in the M&M item that need to be better answered and organized within the text.

- the abstract needs to be rewritten considering the justification of the study, the methods used and the main results obtained. Consider including numerical results as well as more specificities of the study protocol (e.g. number of animals, form and concentration of substance administration and method of fatigue assessment).

- Line#37: some researchers? Only one reference was cited...

- Lines#73 and 74: were these biochemical parameters measured in blood? The abstract must also contain this information.

- Line#88: consider including a definition when bringing abbreviations in the text (only the first time it appears). Check the entire text.

- Line#95: is this the total concentration of the added ingredients? More information about this solution needs to be provided. Furthermore, was this concentration based on prior study? Or in what? You must provide a justification for this.

- Why was the creatine solution used in emulsified form? It is not the most commonly ingested form... Obviously the format of any ingredient in food may or may not increase the availability of ingredients and therefore needs to be described in this item.

- Line#119: gavage is a system that is, in a certain way, stressful for the mouse and therefore can be a stimulator of hormones such as adrenaline that alter carbohydrate metabolism. Is this method the most suitable for this type of study?

- It needs to be clearer in the M&M item what this “coumpound sugar” is.

- Table 1: where are the results for the different groups of mice? I believe that the result is only related to the carbohydrate tolerance test. This needs to be clear in the text.

- Line#190: it is necessary to consider that, as the digestion of carbohydrates begins in the mouth, in this analysis, this is ignored. Consider including a paragraph explaining this issue.

- Line#204: needs to be better explained.

- Line#216: this is information that should be in M&M.

- The entire study was carried out based on the digestion and absorption of carbohydrates, however, no insulin measurement was carried out to confirm the results?

- Line#232: Compared to which compounds?

- Line#255: If the dose used was small compared to what is used in humans (as stated above), how is it possible for this result to be attributed to the dose of creatine? Furthermore, as shown in Figure 4, there is no difference between the creatine group and the others. This demonstrates that any positive or negative result in the time to reach fatigue in mice is probably not due to the presence of creatine.

- Line#261: if there was no difference, it makes no sense to say that it increased or not, especially in studies with animal models where it is assumed that the groups are homogeneous.

- Lines#282-286: this discussion needs to be rewritten because in fact there is only a difference between the muscle glycogen content between the positive treatments and the control.

- Line#292: this statement needs to be rewritten and better targeted as the glycogen content alone does not support it.

- Figure 7: it is necessary to include an explanation of why the lactate content in the control group was not higher than the other treatments, as this would be an expected result.

- Authors need to consider including the ethical statement.

Comments on the Quality of English Language

In general, the manuscript needs to be reviewed for English by a specialist as there are errors that compromise the clarity of the manuscript.

Author Response

Dear Ms. Cecily Gu and Reviewers

We would like to thank the Editor and the Reviewers for your time and insightful comments on our manuscript, referenced above. It has been revised according to your comments. The detailed modifications are listed below point by point. The responses to the comments and suggestions of the reviewer are listed in this document starting on page 2.

Note: Author responses are highlighted in red. All the changes in the revised manuscript have been highlighted in red. All the line numbers in our responses refer to the revised manuscript.

Reviewer #1:

  1. Point-by-point response to Comments and Suggestions for Authors

Comments 1: Although the idea of the study is interesting, I was unsure of the need for the study given that creatine, for example, has already been used in clinical practice for athletes. Therefore, what is the innovation of carrying out a study like this with an animal model instead of an RCT?

Response 1: Thank you for your meaningful comments. The glycemic index (GI) of carbohydrate supplements has an impact on both the glucose level and glycogen content in the body, which can potentially influence exercise endurance and fatigue. However, the effects of compound sugars formulated by different GI sugars on exercise-related fatigue are rarely reported. Simultaneously, the effects of compound sugar combined with creatine emulsion supplementation on exercise ability and exercise-related fatigue need to be further investigated. In this study, we formulated a CS suitable for exercise energy supply, which is composed of different GI sugars. Then the effects of CS and creatine supplementation on exercise ability were explored. About the selection of experimental subjects, we have considered the following two factors. Firstly, this study formulated CS through digestion in vitro and glycemic response, and it necessitated frequent exercise training alongside substrate supplementation. Animal models have been employed for their feasibility in these experimental conditions, remaining compliant with ethical standards for animal research. Secondly, engaging directly in RCTs is often a complex and resource-intensive process that may not yield significantly differential outcomes, potentially resulting in an inefficient use of time and resources. By conducting preliminary investigations using animal models, we aim to lay a foundational framework that will inform the design and implementation of future clinical trials. Additionally, we added the following sentence in the revised manuscript to reflect your helpful suggestion.

Line#384-385

“Moreover, the animal experimental results need to be further verified by randomized controlled trials.”

Comments 2: The abstract needs to be rewritten considering the justification of the study, the methods used and the main results obtained. Consider including numerical results as well as more specificities of the study protocol (e.g. number of animals, form and concentration of substance administration and method of fatigue assessment).

Response 2: Thank you for your helpful suggestion. The abstract has been revised as follows:

Line#10-26

“Abstract: In this study, a compound sugar (CS) with different glycemic index sugar was formulated by hydrolysis characteristics and postprandial glycemic response, and the impact of CS and creatine emulsion on exercise-related fatigue in mice was investigated. 35 C57BL/6 mice were randomly divided into 5 groups to supply different emulsions: initial emulsion (Con), glucose emulsion (62 mg/10 g MW glucose; Glu), CS emulsion (62 mg/10 g MW compound sugar; CS), creatine emulsion (6 mg/10 g MW creatine; Cr), CS and creatine emulsion (62 mg/10 g MW compound sugar, 6 mg/10 g MW creatine, CS-Cr), for 4 weeks. Then the exhaustion time of weight-bearing swimming and forelimb grip strength were measured to evaluate the exercise capacity of mice and some fatigue-related biochemical indexes of blood were determined. The results demonstrated that the ingestion of CS significantly reduced the peak of postprandial blood glucose levels and prolonged the energy supply of mice as compared to ingesting an equal amount of glucose. Mouse exhaustion time was 1.22-fold longer in the CS group than in the glucose group. Additionally, the supplementation of CS increased the liver glycogen content and total antioxidant capacity of mice. Moreover, the combined supplementation of CS and creatine increased relative forelimb grip strength and decreased blood creatine kinase activity. The findings suggested that the intake of CS could enhance exercise capacity, and the combined supplementation of CS and creatine has a synergistic effect in improving performance.”

Comments 3: Line#37: some researchers? Only one reference was cited...

Response 3: Thank you for your comment. We have supplemented reference [7] in the revised version (Line#43) and revised the citations of the full-text references based on your suggestion.

Reference

  1. Burke, L.M.; Hawley, J.A.; Wong, S.H.S.; Jeukendrup, A.E. Carbohydrates for training and competition. J. Sports Sci. 2011, 29, S17-S27. https://doi.org/10.1080/02640414.2011.585473

Comments 4: Lines#73 and 74: were these biochemical parameters measured in blood? The abstract must also contain this information.

Response 4: Thank you for your comment. Yes, these biochemical parameters were measured in blood. We have added this information to the revised manuscript and abstract based on your suggestion.

Line#78-81

“Through the determination of glycogen contents and biochemical indexes in the blood such as lactate, blood urea nitrogen (BUN), creatinine, creatine kinase (CK) and total antioxidant capacity (T-AOC), the anti-fatigue effects of CS and creatine were compared.”

Comments 5: Line#88: consider including a definition when bringing abbreviations in the text (only the first time it appears). Check the entire text.

Response 5: Thank you for your suggestion. We have checked the entire text. As suggested by the reviewer, we have corrected the “SPF” to “specific pathogen-free”.

Line#93-95

“The mice were housed in separate cages in the specific pathogen-free environment for adaptive rearing for the first week when they were provided with clean water and basic feed.”

Comments 6: Line#95: is this the total concentration of the added ingredients? More information about this solution needs to be provided. Furthermore, was this concentration based on prior study? Or in what? You must provide a justification for this.

Response 6: Thank you for your comment. Yes, this concentration was selected based on the previous study [14] and the stability of the emulsion formed by this method has been verified. We have revised the manuscript to reflect your comment as follows:

Line#100-103

“The preparation method of the emulsion was referred to in a previous study with some modifications [14]. The WPI solution with a concentration of 10mg/mL was pre-pared and mixed with various compounds (i.e., 250 mg/ mL glucose, 250 mg/ mL CS and 25 mg/mL creatine) [7,15].”

References

  1. Gao, T.T.; Liu, J.X.; Gao, X.; Zhang, G.Q.; Tang, X.Z. Stability and digestive properties of a dual-protein emulsion system based on soy protein isolate and whey protein isolate. Foods 2023, 12, https://doi.org/10.3390/foods12112247
  2. Burke, L.M.; Hawley, J.A.; Wong, S.H.S.; Jeukendrup, A.E. Carbohydrates for training and competition. J. Sports Sci. 2011, 29, S17-S27. https://doi.org/10.1080/02640414.2011.585473
  3. Kreider, R.B.; Kalman, D.S.; Antonio, J.; Ziegenfuss, T.N.; Wildman, R.; Collins, R.; Candow, D.G.; Kleiner, S.M.; Almada, A.L.; Lopez, H.L. International society of sports nutrition position stand: safety and efficacy of creatine supplementation in exercise, sport, and medicine. J. Int. Soc. Sports Nutr. 2017, 14, 18-36. https://doi.org/10.1186/s12970-017-0173-z

Comments 7: Why was the creatine solution used in emulsified form? It is not the most commonly ingested form… Obviously the format of any ingredient in food may or may not increase the availability of ingredients and therefore needs to be described in this item.

Response 7: Thank you for your meaningful comments. Due to the low solubility of creatine, the common form of creatine products is solid beverages. We applied creatine to the emulsion form which could provide a new form of creatine supplement. In addition, water-soluble and fat-soluble nutrients can be added to the emulsion, which can make the nutrients of creatine supplements more comprehensive. It remains valuable for us to explore whether the form of the emulsion affects the availability of ingredients in future experiments.

Comments 8: Line#119: gavage is a system that is, in a certain way, stressful for the mouse and therefore can be a stimulator of hormones such as adrenaline that alter carbohydrate metabolism. Is this method the most suitable for this type of study?

Response 8: Thank you for your comment. The determination of postprandial blood glucose response requires mice to ingest the same amount of carbohydrates and determine the blood glucose at designated time intervals after the ingestion. Free feeding is relatively the least likely method to cause tension or stress in mice, but it is difficult to control food intake and eating time. Therefore, taking the same amount of intragastric administration is the most suitable method for this study. Moreover, many studies have used the method of intragastric administration to determine glycemic response. Specific references are listed as follows.

References

  1. Hung, P.V.; Chau, H.T.; Lan Phi, N.T. In vitro digestibility and in vivo glucose response of native and physically modified rice starches varying amylose contents. Food Chem. 2016, 191, 74-80. https://doi.org/10.1016/j.foodchem.2015.02.118
  2. 1. Ryu, J.J.; Li, X.L.; Lee, E.S.; Li, D.; Lee, B.H. Slowly digestible property of highly branched α-limit dextrins produced by 4,6-α-glucanotransferase from Streptococcus thermophilus evaluated in vitro and in vivo. Carbohydr. Polym. 2022, 275, https://doi.org/10.1016/j.carbpol.2021.118685
  3. Guo, J.Y.; Ellis, A.; Zhang, Y.Q.; Kong, L.Y.; Tan, L.B. Starch-ascorbyl palmitate inclusion complex, a type 5 resistant starch, reduced in vitro digestibility and improved in vivo glycemic response in mice. Carbohydr. Polym. 2023, 321, https://doi.org/10.1016/j.carbpol.2023.121289
  4. Liu, J.; Wang, M.Z.; Peng, S.L.; Zhang, G.Y. Effect of green tea catechins on the postprandial glycemic response to starches differing in amylose content. J. Agric. Food. Chem. 2011, 59, 4582-4588. https://doi.org/10.1021/jf200355q

Comments 9: It needs to be clearer in the M&M item what this “compound sugar” is.

Response 9: Thank you for your comment. Here compound sugar is the CS mentioned earlier, and we have revised all the compound sugar in the M&M item to CS.

Line#139-143

” (3) CS group was treated with CS emulsion (62 mg/10 g MW CS); (4) Creatine group was treated with creatine emulsion (6 mg/10 g MW creatine; Cr); (5) CS and creatine combination group was treated with CS and creatine emulsion (62 mg/10 g MW CS, 6 mg/10 g MW creatine, CS-Cr).”

Comments 10: Table 1: where are the results for the different groups of mice? I believe that the result is only related to the carbohydrate tolerance test. This needs to be clear in the text.

Response 10: I am sorry for the inconvenience caused to you because of the unclear expression. Mice's small intestine extracts all from the normal mice that did not receive any treatment. The result is only related to hydrolytic properties and glycemic responses. We have revised the manuscript to reflect your suggestion as follows:

Line#112-114

“Firstly, the healthy mice that did not receive any treatment were killed by neck dissection after 16 h of fasting without water, and the small intestinal parts were dissected immediately. “

Comments 11: Line#190: it is necessary to consider that, as the digestion of carbohydrates begins in the mouth, in this analysis, this is ignored. Consider including a paragraph explaining this issue.

Response 11: Thank you for your comment. Carbohydrate digestion is mainly divided into two stages, the first is the oral cavity, in which the most prominent enzyme is α-amylase, which cleaves α-(1-4) glycosidic bonds to decompose starch into soluble maltose and glucan [21]. The second stage occurs in the upper small intestine. Sugars in this study were hardly digested and hydrolyzed in the mouth, so the digestion of sugars did not begin in the mouth. We have added an explanation of this issue to the manuscript based on your suggestion.

Line#191-193

“The main digestion and absorption sites of disaccharides and oligosaccharides are in the small intestine [21]. To determine the hydrolytic properties, MISE was used directly to analyze the amounts of glucose released from various sugars (Figure 1).”

Reference

  1. Pedersen, A.M.L.; Sorensen, C.E.; Proctor, G.B.; Carpenter, G.H. Salivary functions in mastication, taste and textural perception, swallowing and initial digestion. Oral Dis. 2018, 24, 1399-1416. https://doi.org/10.1111/odi.12867

Comments 12: Line#204: needs to be better explained.

Response 12: Thank you for your comment. We have rewritten this part according to your suggestion:

 Line#219-222

“These results were comparable to those of a previous human experiment that isomaltulose reduced the increase in postprandial blood glucose and insulin concentration compared with sucrose [23].”

Reference

  1. Van Can, J.G.P.; Van Loon, L.J.C.; Brouns, F.; Blaak, E.E. Reduced glycaemic and insulinaemic responses following trehalose and isomaltulose ingestion: Implications for postprandial substrate use in impaired glucose-tolerant subjects. Br. J. Nutr. 2012, 108, 1210-1217. https://doi.org/10.1017/s0007114511006714

Comments 13: Line#216: This is information that should be in M&M.

Response 13: Thank you for your helpful suggestion. The information has been revised to M&M:

Line#103-104

“CS was composed of glucose, maltodextrin, sucrose, and isomaltose with a mass ratio of 4: 5 : 6: 15.”

Comments 14: The entire study was carried out based on the digestion and absorption of carbohydrates, however, no insulin measurement was carried out to confirm the results?

Response 14: Thank you for this comment, and we appreciate your suggestion. We agree that more studies would be useful to confirm the results and we will consider this point in follow-up research. In our study, the mice we used were healthy and their ability of carbohydrate metabolism was normal. And we only measured the blood sugar response referring to previous research [18]. In addition, according to the experimental results of Judith et al [23], after ingestion of carbohydrates, the changing trend of insulin level is similar to that of blood glucose.

References

  1. Hung, P.V.; Chau, H.T.; Lan Phi, N.T. In vitro digestibility and in vivo glucose response of native and physically modified rice starches varying amylose contents. Food Chem. 2016, 191, 74-80. https://doi.org/10.1016/j.foodchem.2015.02.118
  2. Van Can, J.G.P.; Van Loon, L.J.C.; Brouns, F.; Blaak, E.E. Reduced glycaemic and insulinaemic responses following trehalose and isomaltulose ingestion: Implications for postprandial substrate use in impaired glucose-tolerant subjects. Br. J. Nutr. 2012, 108, 1210-1217. https://doi.org/10.1017/s0007114511006714

Comments 15: Line#232: Compared to which compounds?

Response 15: Thank you very much for pointing out this mistake. We have revised the manuscript based on your suggestion:

Line#248-250

The results showed that CS could achieve better maintenance of glucose concentrations and provide more stable carbohydrate availability during exercise compared to glucose [7].”

Comments 16: Line#255: If the dose used was small compared to what is used in humans (as stated above), how is it possible for this result to be attributed to the dose of creatine? Furthermore, as shown in Figure 4, there is no difference between the creatine group and the others. This demonstrates that any positive or negative result in the time to reach fatigue in mice is probably not due to the presence of creatine.

Response 16: Thank you for your comments and it means a lot to us. The daily recommended amount of creatine for a 60 kg human is 3 g [15]. The dose for mice is converted according to the body surface area, and the conversion coefficient is 12.3 [26]. Athletes tend to supplement higher levels of creatine than the recommended dose, so that the content of creatine in the body is relatively saturated as soon as possible, and exercise training will accelerate the loss of creatine [38]. We have revised the manuscript based on your comments:

Line#266-272

“There were no significant differences between the Glu group and the Con group. However, compared to those in the glucose group, mice supplemented with CS demonstrated a significant increase in their average exhaustion time compared to glucose (The exhaustion time of CS and CS-Cr was 1.22 and 1.18 times that of Glu, respectively). Moreover, there was no significant difference between CS and CS-Cr. These results demonstrated a significant improvement in mice’s exercise endurance when supplied with CS.”

References

  1. Kreider, R.B.; Kalman, D.S.; Antonio, J.; Ziegenfuss, T.N.; Wildman, R.; Collins, R.; Candow, D.G.; Kleiner, S.M.; Almada, A.L.; Lopez, H.L. International society of sports nutrition position stand: safety and efficacy of creatine supplementation in exercise, sport, and medicine. J. Int. Soc. Sports Nutr. 2017, 14, 18-36. https://doi.org/10.1186/s12970-017-0173-z
  2. Lee, M.C.; Hsu, Y.J.; Yang, L.H.; Huang, C.C.; Ho, C.S. Ergogenic effects of green tea combined with isolated soy protein on increasing muscle mass and exercise performance in resistance-trained mice. Nutrients 2021, 13, 4547. https://doi.org/10.3390/nu13124547
  3. Kreider, R.B.; Jager, R.; Purpura, M. Bioavailability, efficacy, safety, and regulatory status of creatine and related compounds: A critical review. Nutrients 2022, 14, 1035-1086. https://doi.org/10.3390/nu14051035

Comments 17: Line#261: if there was no difference, it makes no sense to say that it increased or not, especially in studies with animal models where it is assumed that the groups are homogeneous.

Response 17: Thank you for your meaningful comment. We have revised the manuscript based on your comment:

Line#277-278

“The forelimb grip of mice was shown in Figure 5(A), and there was no significant difference among groups.”

Comments 18: Lines#282-286: this discussion needs to be rewritten because in fact there is only a difference between the muscle glycogen content between the positive treatments and the control.

Response 18: Thank you for your comment. We have revised the manuscript based on your suggestion:

Line#298-301

“The muscle glycogen content of the mice supplemented with glucose, creatine and CS was significantly higher compared to the blank control group, with no significant difference among the four groups (Figure 6 (A)).”

Comments 19: Line#292: this statement needs to be rewritten and better targeted as the glycogen content alone does not support it.

Response 19: Thank you for your suggestion. We have rewritten this statement based on your suggestion:

Line#306-308

“These results suggested that CS supplementation could raise the content of glycogen in mice, and combined supplementation with CS and creatine could additionally enhance liver glycogen content.”

Comments 20: Figure 7: it is necessary to include an explanation of why the lactate content in the control group was not higher than the other treatments, as this would be an expected result.

Response 20: Thank you for your helpful suggestion. The explanation has been added as follows:

Line#321-324

“Figure 7 (A) illustrates that no significant variation in lactate content was observed among the different groups. This suggested that neither CS nor creatine could prevent muscle injury by reducing the production of lactate during anaerobic exercise [30]."

References

  1. Feng, Z.Y.; Wei, Y.; Xu, Y.G.; Zhang, R.X.; Li, M.L.; Qin, H.M.; Gu, R.Z.; Cai, M.Y. The anti-fatigue activity of corn peptides and their effect on gut bacteria. J. Sci. Food Agric. 2022, 102, 3456-3466. https://doi.org/10.1002/jsfa.11693

Comments 21: Authors need to consider including the ethical statement.

Response 21: Thank you for your suggestion. The ethical statement has been added to the revised manuscript as follows:

Line#397-400

Ethical Statement: The Ethics Committee of the Laboratory Animal Center of Jiangnan University (Permission number: JN.No20220415c0500620[128]) approved all experiments, which were conducted by the Guide for the Care and Use of Laboratory Animals (8th edition, National Academy of Sciences Press).”

  1. Response to Comments on the Quality of English Language

Comments: In general, the manuscript needs to be reviewed for English by a specialist as there are errors that compromise the clarity of the manuscript.

Response: Thank you for your suggestion. We have invited a specialist to help polish our manuscript. These changes will not influence the content and framework of the paper. We marked them in red in the revised manuscript. We appreciate for Reviewers warm work earnestly and hope that the correction will meet with approval.

-The end of the Response List.

Once again, we appreciate all the suggestions and queries raised by all the reviewers, which we believe have helped us to modify the manuscript and to improve the quality of the revised paper.

Sincerely Yours,

Yue Li

State Key Laboratory of Food Science and Resources

School of Food Science and Technology

Jiangnan University

Wuxi, Jiangsu, 214122, Jiangsu Province, China

Reviewer 2 Report

Comments and Suggestions for Authors

COMMENTS TO AUTHORS

Carbohydrate consumption is known to improve exercise performance by preventing metabolite buildup and extending time to exhaustion. The results on which carbohydrates (low glycemic index (GI) or high GI) and whether to combine them with other performance enhancers such as creatine, has remained to be determined. Liao et al sought to investigate whether glucose or a compound sugar (CS) of glucose, maltodextrin, sucrose, and isomaltose in combination with creatine (Cr) could improve exercise performance in mice. The authors found that CS and the CS+Cr combination were the most effective at improving their chosen markers of exercise performance. The manuscript addresses an interesting and important question in the field of exercise performance and the results are relevant for the field. However, I believe there are three important control groups missing before a definitive statement can be made on the superiority of CS on exercise outcomes. I suggest the authors revise and resubmit.

Major points

1.     With the current experimental groups, the authors may conclude that CS supplementation is superior to Glu supplementation on exercise performance. However, it remains to be determined whether Glu+Cr would similarly improve exercise. Cr itself improved exercise outcomes, so it is possible that Glu+Cr may also exhibit improvements and the results are not CS specific. Therefore, a Glu+Cr control group must be run.

2.     The authors should also consider including low GI and low GI+Cr groups as well. In the introduction, the authors discuss the efficacy of low GI carbohydrates, but then proceed to not test a low GI carb individually. I suggest repeating the experiments with two additional groups: one receiving isomaltose and another receiving isomaltose + Cr. The present results don’t address whether CS allows for better exercise performance than isomaltose. The present results only show that it performs better than the high GI, glucose.

3.     Figure 1, 2, and 3 should be analyed using a repeated measures ANOVA not a one way ANOVA as the data is being collected from the same mice for each time point.

Minor points

1.     Provide markers of significance on Figures 1, 2, and 3 to visualize which time points are significantly different among/between the groups.

2.     Line 35: “endurance” should be changed to “increases”.

3.     Line 52: “et al” should be added after “Rolando”.

4.     Move Lines 215-217 in the methods and describe the CS formulation in more detail in the methods section.

Author Response

Dear Editors and Reviewers

We would like to thank the Editor and the Reviewers for your time and insightful comments on our manuscript, referenced above. It has been revised according to your comments. The detailed modifications are listed below point by point. The responses to the comments and suggestions of the reviewer are listed in this document starting on page 2.

Note: Author responses are highlighted in red. All the changes in the revised manuscript have been highlighted in red. All the line numbers in our responses refer to the revised manuscript.

Reviewer #2:

  1. Point-by-point response to Comments and Suggestions for Authors

Comments 1: With the current experimental groups, the authors may conclude that CS supplementation is superior to Glu supplementation on exercise performance. However, it remains to be determined whether Glu+Cr would similarly improve exercise. Cr itself improved exercise outcomes, so it is possible that Glu+Cr may also exhibit improvements and the results are not CS specific. Therefore, a Glu+Cr control group must be run.

Response 1: Thank you for your suggestion and it means a lot to us. At present, glucose is the main sugar supplement for athletes. In our study, we established the Glu group, along with the blank control group and CS group, to compare the efficacy of CS versus traditional sugar in exercise. The results demonstrated that CS outperforms glucose supplementation. In addition, we established the Cr control group and Cr+CS group to investigate whether the combined supplementation of CS and creatine yields significantly superior effects compared to CS or creatine alone. These five groups fulfilled the research objectives of our study. Due to the extended duration of the Glu + Cr group experiment and the potential for significant errors between the two experiments, we decided not to implement this modification. Nevertheless, it remains valuable for us to establish a Glu+Cr control group in future experiments, allowing further exploration of whether the combined effect of CS and creatine supplementation surpasses that of the glucose + creatine combination.

Comments 2: The authors should also consider including low GI and low GI+Cr groups as well. In the introduction, the authors discuss the efficacy of low GI carbohydrates, but then proceed to not test a low GI carb individually. I suggest repeating the experiments with two additional groups: one receiving isomaltose and another receiving isomaltose + Cr. The present results don’t address whether CS allows for better exercise performance than isomaltose. The present results only show that it performs better than the high GI, glucose.

Response 2: Thank you for your helpful suggestion and it means a lot to us. Despite the slow energy supply rate of low glycemic index (GI) sugars like isomaltose, which may impact sports performance as it could result in untimely energy provision, regulating this combination can enhance the scientific rigor and credibility of the research. We will consider this point in follow-up research. Additionally, we added the following sentence in the revised manuscript to reflect your helpful suggestion.

Line#381-384

“Further studies are required to investigate the effects of CS-Cr compared to glucose or low GI sugars (i.e., isomaltulose) with creatine.”

Comments 3: Figure 1, 2, and 3 should be analyzed using a repeated measures ANOVA not a one-way ANOVA as the data is being collected from the same mice for each time point.

Response 3: Thank you for your reminder. We have used repeated measures ANOVA to analyze Figures 1, 2, and 3 as suggested and have revised the manuscript to reflect your comment as follows:

Line#168-172

“Data were expressed as mean ± standard deviation (SD). The result of hydrolytic properties and glycemic responses were analyzed by repeated measures analysis of variance (ANOVA) with SPSS (version 20.0; SPSS, Chicago, IL, USA), and the other data were analyzed by one-way ANOVA. A Duncan’s test was used to measure the difference between different treatments. P < 0.05 was taken as the standard of statistical significance.”

Comments 4: Provide markers of significance on Figures 1, 2, and 3 to visualize which time points are significantly different among/between the groups.

Response 4: Thank you for your suggestion. We have added markers of significance in Figures 1, 2, and 3 based on your suggestions. The improved Figures are shown as follows:

Figure 1. Hydrolytic properties of different types of sugar (1 g/L) for 24 h by mice small intestine extract as a source of mammalian α-glucosidases. Data are expressed as means ± SD (n = 3). Different capital letters indicate significant differences between time points in the same group, and different lower-case letters indicate significant differences between groups at the same time point (p < 0.05).

Figure 2. Glycemic responses after the ingestion of individual sugars. Data are expressed as means ± SD (n = 7). Different capital letters indicate significant differences between time points in the same group, and different lower-case letters indicate significant differences between groups at the same time point (p < 0.05).

Figure 3. Glycemic responses after the ingestion of compound sugar. Data are expressed as means ± SD (n = 7). Different capital letters indicate significant differences between time points in the same group, and different lower-case letters indicate significant differences between groups at the same time point (p < 0.05).

Comments 5: Line 35: “endurance” should be changed to “increases”.

Response 5: We were sorry for our careless mistakes. Thank you for your reminder. This information has been corrected in the revised manuscript based on your suggestions.

Line#39-42

“Furthermore, the glycemic index (GI) of carbohydrates may influence endurance during exercise. It was reported that substituting sucrose with isomaltose (a low GI disaccharide) significantly increases the exercise endurance of mice for four weeks [5].”

Comments 6: Line 52: “et al” should be added after “Rolando”.

Response 6: Thank you for your comment. This information has been added to the revised manuscript based on your suggestions.

Line#58-60

“According to Ceddia et al., supplementing with creatine might increase glucose oxidation in L6 rat skeletal muscle cells while reducing lactate production [10].”

Comments 7: Move Lines 215-217 in the methods and describe the CS formulation in more detail in the methods section.

Response 7: Thank you for your suggestion. The method has been revised as follows:

Line#101-104

“The WPI solution with a concentration of 10mg/mL was prepared and mixed with various compounds (i.e., 250 mg/ mL glucose, 250 mg/ mL CS and 25 mg/mL creatine) [7,15]. CS was composed of glucose, maltodextrin, sucrose, and isomaltose with a mass ratio of 4: 5 : 6: 15.”

References

  1. Burke, L.M.; Hawley, J.A.; Wong, S.H.S.; Jeukendrup, A.E. Carbohydrates for training and competition. J. Sports Sci. 2011, 29, S17-S27. https://doi.org/10.1080/02640414.2011.585473
  2. Kreider, R.B.; Kalman, D.S.; Antonio, J.; Ziegenfuss, T.N.; Wildman, R.; Collins, R.; Candow, D.G.; Kleiner, S.M.; Almada, A.L.; Lopez, H.L. International society of sports nutrition position stand: safety and efficacy of creatine supplementation in exercise, sport, and medicine. J. Int. Soc. Sports Nutr. 2017, 14, 18-36. https://doi.org/10.1186/s12970-017-0173-z

-The end of the Response List.

Once again, we appreciate all the suggestions and queries raised by all the reviewers, which we believe have helped us to modify the manuscript and to improve the quality of the revised paper.

Sincerely Yours,

Yue Li

State Key Laboratory of Food Science and Resources

School of Food Science and Technology

Jiangnan University

Wuxi, Jiangsu, 214122, Jiangsu Province, China

Round 2

Reviewer 1 Report

Comments and Suggestions for Authors

Dear editor,

The effort the authors made to respond to the suggestions this reviewer made to improve the quality of the manuscript is admirable. The manuscript seems much more compatible with the standard of this journal and in my opinion it can be accepted for publication.

Reviewer 2 Report

Comments and Suggestions for Authors

I appreciate the authors responding to my suggestions.